# Changes in Opioid Prescribing Behaviors among Family Physicians Who Participated in a Weekly Tele-Mentoring Program

**DOI:** 10.3390/jcm9010014

**Published:** 2019-12-19

**Authors:** Santana Díaz, Jane Zhao, Shawna Cronin, Susan Jaglal, Claire Bombardier, Andrea D. Furlan

**Affiliations:** 1Institute of Medical Sciences, University of Toronto, Toronto, ON M5S, Canada; 2Toronto Rehabilitation Institute, University Health Network, Toronto, ON M5G1L7, Canada; 3Institute of Health Policy, Management and Evaluation, University of Toronto, Toronto, ON M5S, Canada; 4Department of Physical Therapy, University of Toronto, Toronto, ON M5S, Canada; 5Department of Medicine, University of Toronto, Toronto, ON M5S, Canada; 6Mount Sinai Hospital, Toronto, ON M5G1X5, Canada; 7Institute for Work & Health, Toronto, ON M5G2E9, Canada

**Keywords:** opioid, chronic pain, tele-mentoring, observational study, controlled study

## Abstract

A weekly tele-mentoring program was implemented in Ontario to help address the growing opioid crisis through teaching and mentoring family physicians on the management of chronic pain and opioid prescribing. This study assessed opioid prescribing behaviours among family physicians who attended the tele-mentoring program compared to two groups of Ontario family physicians who did not attend the program. We conducted a retrospective cohort study with two control groups: a matched cohort, and a random sample of 3000 family physicians in Ontario. Each physician was followed from one year before the program, which is the index date, and one year after. We examined the number and proportion of patients on any opioid, on high dose opioids, and the average daily morphine equivalent doses prescribed to each patient. We included 24 physicians who participated in the program (2760 patients), 96 matched physicians (11,117 patients) and 3000 random family doctors (374,174 patients). We found that, at baseline, the tele-mentoring group had similar number of patients on any opioid, but more patients on high dose opioids than both control groups. There was no change in the number of patients on any opioid before and after the index date, but there was a significant reduction in high-dose opioid prescriptions in the extension for community healthcare outcomes (ECHO) group, compared to a non-significant increase in the matched cohort, and a non-significant reduction in the Ontario group during the same comparable periods. Participation in the program was associated with a greater reduction in high-dose opioid prescribing.

## 1. Introduction

Chronic pain is estimated to affect 19% of Canadians, more than half of whom report that their pain has persisted for more than 10 years [1]. In Canada, most patients with chronic pain are managed by family physicians, however, family physicians receive little training in managing this complex condition [2,3]. North America has been facing an opioid crisis that started around 1999. High rates of opioid prescribing by physicians have been implicated in the first phase of the opioid crisis [4]. In 2014/15, one in seven Ontarians (i.e., 1.6 million people) were dispensed an opioid, accounting for a total of 9 million dispensing events in a year [5].

Medical education is a strategy used to improve opioid prescribing among physicians [6]. Tele-mentoring educational programs are a novel approach to bringing knowledge to remote and underserved settings. In recent years, these programs have expanded in Canada [7] and around the world [8]. Project Extension for Community Healthcare Outcomes (ECHO) is a tele-mentoring program that started with the need to provide care to hepatitis C patients in rural areas of New Mexico [9]. The ECHO model provides access to specialty treatment in rural and underserved areas by providing front-line clinicians, such as family physicians, with the knowledge and support they need to manage patients with complex conditions. [9]. There are four pillars to the ECHO model: the utilization of video technology, adherence to best practices to reduce variations in care, case-based learning, and monitoring outcomes. In 2014, the first ECHO program was launched in Canada [7]. ECHO Ontario Chronic Pain/Opioid Stewardship (ECHO) aimed to disseminate knowledge and enhance the capacity of family physicians to improve the management of patients with chronic pain and reduce opioid-related problems. Our group demonstrated that participation in ECHO led to a significant increase in chronic pain self-efficacy and knowledge, however, these were self-reported results, and there is a need for objective measures of skills and competence [10]. ECHO continues to be offered as an ongoing program, free of cost, to healthcare providers in Ontario (www.EchoOntario.ca).

For this study, we used an existing database of narcotics prescriptions to compare the opioid-prescribing behaviors of family physicians before and after attending ECHO with those who did not attend ECHO in Ontario.

## 2. Methods

### 2.1. Intervention

ECHO Ontario Chronic Pain/Opioid Stewardship (ECHO) started in June 2014, and offered weekly 2 h sessions that started with a 20 min didactic on a topic given by a member of the interdisciplinary team (“the hub”) or invited speaker, followed by one or more de-identified patient case discussions presented by one of the primary care providers who is registered to attend ECHO (“the spoke”). For each case presented, spokes receive suggestions about differential diagnoses, additional investigations, non-pharmacological and pharmacological approaches and lifestyle modifications. There are no costs to participate in ECHO for any participant.

### 2.2. Study Design

We employed an observational study design using a retrospective time series with two control groups. The study was approved by the Research Ethics Boards at UHN and the University of Toronto. Health administrative databases held at Institute for Clinical Evaluative Sciences (ICES) were used to identify control groups and conduct analyses; ICES is an independent, non-profit research institute funded by an annual grant from the Ontario Ministry of Health and Long-Term Care (MOHLTC). As a prescribed entity under Ontario’s privacy legislation, ICES is authorized to collect and use health care data for the purposes of health system analysis, evaluation and decision support. Secure access to these data is governed by policies and procedures that are approved by the Information and Privacy Commissioner of Ontario.

### 2.3. Inclusion and Exclusion Criteria

The ECHO attendance database was used to identify potential participants. The data regarding prescriptions were obtained using the Narcotics Monitoring System (NMS) at ICES. Participants were contacted to give consent to share their names with ICES [11].

We included three groups of Ontario family physicians:Study group: physicians who participated in the ECHO Ontario Chronic Pain/Opioid Stewardship from June 2014 to December 2016 and who attended a minimum of two sessions;Control group 1: matched 4:1 using a greedy matching technique [12] based on sex, type of practice, location rural or urban, and years in practice. To ensure the controls had at least some prescribing throughout the window, the first and last date of prescriptions within the window were identified. Most (>90%) of the physicians prescribed regularly within the five year period, and these were included in the matching cohort;Control group 2: all Ontario Family Physicians (total *n* = 8493) with at least one opioid prescription spanning the two year observation window of the study. However, due to the large number of family physicians in Ontario, a random sample of 3000 Ontario physicians were identified for analysis;Exclusions:Physicians who were not eligible to bill for a full year before the ECHO start date;Physicians who did not prescribe any opioids during the two-year observation window;Physicians with missing variables for matching;Physicians who practiced outside of Ontario during the study period;Physicians who were residents or medical students;

At the level of the patients: those who did not receive an opioid prescription before the index start date were not included in the analyses.

### 2.4. Administrative Data Sources

We collected basic demographics data from the physicians, such as: type of practice (solo/group practice), number of years in practice, number of ECHO sessions attended, and first date of ECHO attendance. ICES linked the physicians with the Corporate Provider Database (CPDB) and the NMS database, then grouped the pre-ECHO and post-ECHO data according to the period of time the physicians attended ECHO for. The ICES physician database (IPDB) contains information about physicians’ demographics, such as: year of birth, sex, country of medical school, year of MD graduation, main practice specialty, and total consultations [13].

The narcotics monitoring system (NMS) has been collecting pharmacy-level dispensing data of all controlled substances (opioids, benzodiazepines, barbiturates, and amphetamines) in Ontario since 2012 [14]. The NMS does not capture opioid prescriptions that are filled in hospitals or in prisons. The NMS collects dispensing data regardless of whether the prescription is paid for under a publicly-funded drug program, through private insurance, or by cash. The NMS has information such as patients’ birth date, drug identification number (DIN), the licensing college of the prescriber (physician or nurse practitioner), quantity of the drug dispensed, and strength of drug in a category/class [14]. We created a DIN list for our study to identify only opioids from the NMS database. Only aggregated data were shared throughout the study period. These datasets were linked using unique encoded identifiers and analyzed at ICES.

### 2.5. Recruitment and Groups Creation

Between June 2014 and December 2016, 93 physicians participated in ECHO; of those, 26 did not meet the criteria for the following reasons: did not attend minimum of two ECHO sessions during the study period, moved outside of Ontario during the study period, or were in residency.

The 67 eligible family physicians were invited by e-mail, followed by two reminders. Of the 30 physicians who responded and showed interest, 26 returned the signed consent form, leaving a sample of 26 physicians (39% response rate). After transfer and linkage to an administrative database, two physicians had missing information for sex and had to be excluded from the rest of the analyses because it was impossible to match them to control physicians, leaving a total of 24 ECHO physicians included in the final analyses.

For these 24 physicians, we pulled information regarding opioid prescriptions one year before and one year after the ECHO starting date. The matched control cohort was created by first pulling all prescribers from the NMS between 1 January, 2013 and 31 December, 2017, a time span which includes the ECHO start dates, as well as the years before and after, to ensure the controls have prescribed within the same period. These were linked with the IPDB and CPDB to obtain other physician characteristics and variables for matching. We then created a cohort of Ontario primary care physicians by taking a simple random sample of 3000 from all comprehensive primary care physicians in Ontario. We only used physicians considered to be comprehensive primary care physicians, identified by an ICES algorithm [15], to avoid including family physicians with a focused practice (e.g., palliative care, chronic pain or addiction medicine). In our study, we included buprenorphine patch and buprenorphine/naloxone sublingual tablets, but not methadone. The dataset of all physicians was then used to identify prescriptions from the NMS between one year prior and one year post the ECHO start date (or index date for the control groups).

### 2.6. Outcome Variables

The primary outcomes were the number and proportion of patients on opioids and high dose of opioids per physician. First, we identified the number of patients who were prescribed opioids at any point within each of four six-month time intervals (12–6 months pre-ECHO, 6–0 months pre-ECHO, 0–6 months post-ECHO, and 6–12 months post-ECHO) per physician. The proportion of patients who were prescribed a high dose was calculated. High dose was defined as dispensed opioids equal or above 200 mg of morphine equivalent doses (MED) per day according to the 2010 Canadian Opioid Guideline [16]. We calculated MEDs using NMS data and the Ontario Prescription Opioid Tool for calculation of conversions from one opioid to another [17]. Patients who were prescribed fentanyl at any point during the two-year observation window were flagged to allow for further analyses of this subgroup.

### 2.7. Data Analyses

First, we compared the baseline characteristics of ECHO physicians who consented to data linkage to those who did not consent to data linkage using data collected in the ECHO attendance database. We used descriptive statistics to compare sex, rurality, years in practice, and geographical location of practice. These analyses were completed at the Toronto Rehabilitation Institute using Microsoft Excel^®^.

To describe the cohorts of physicians identified through health administrative databases, we used descriptive statistics to compare basic demographic and clinical characteristics between the ECHO physicians, matched controls, and the Ontario random sample. We used one-way ANOVA to examine the difference in mean proportion of patients taking opioids and high-dose opioids by physician, comparing between the ECHO physicians, matched controls, and Ontario at each time interval. Next, we took the difference in the proportion of patients on a high dose and used one-way ANOVA to identify whether the change in proportions was significantly different across the groups.

Due to the large amount of opioid prescriptions during the study period, we analyzed patient level data by calculating the mean monthly MED for the six months before and six months after the corresponding physicians’ index date (not for 12 months before and after). A subset of the patients who had a MED >200 at some point were analyzed in the same way to identify significant prescription changes in this group.

To assess changes in the pre/post periods between the groups, we used one-way ANOVAs for six months pre/post, and 12 months pre/post periods. Standardized differences were also calculated to quantify the effect size between (a) the ECHO group and the matched control, and (b) between the ECHO group and the Ontario physicians. All data analyses were conducted in SAS software version 11 (SAS Institute) considering type 1 error equal to 5%.

## 3. Results

There were 116 ECHO sessions during the study period (June 2014 to December 2016). The topics of the short didactics included: the five pillars of chronic pain, goals setting, pain assessment, sleep, opioid guidelines, opioid switching, opioid tapering, cannabinoids, urine drug testing, exercises and physical modalities, psychological approaches, and specific chronic pain conditions, including back pain, myofascial pain, neuropathic pain, central sensitization, fibromyalgia, post-surgical pain and phantom limb pain. A total of 5594 h of medical education were provided and 2797 Continuing Medical Education (CME) hours were granted. Three in-person weekend workshops were also offered.

### 3.1. Characteristics of Physicians from ECHO

The 26 physicians who gave consent to participate attended on average 17 (SD = 10.8) ECHO sessions. Of the 24 physicians who were included in the ICES analyses, there were 14 female (58.3%) and 10 male (41.7%); 18 (75%) worked in Family Health Teams, and six (25%) worked in rural locations.

### 3.2. Differences between Participants and Non-Participants

The physicians who did not consent to data linkage were similar to the 24 physicians that were included in the final analyses in relation to sex (66.6% female), rurality (30%) and years in practice. There were small but non-significant differences in relation to geographical location (*p* = 0.881) and practice type (*p* = 0.157). In terms of practice type, the physicians who agreed were mostly from Family Health Teams (75%), while the physicians who did not agree were mostly in Community Health Centres (only 33% were from Family Health Teams).

### 3.3. Differences and Similarities across the Three Cohorts

The matched cohort was very similar to the ECHO group in relation to the variables used for the matching, and for other variables that were not matched (geographical location and physicians’ age). When compared to the Ontario physicians, the mean age in the ECHO group was significantly younger than Ontario physicians (45.89 vs 52.23 years), predominantly female (58.3% vs 46.4%), with more representation from rural communities (25% vs 7.6%) and working predominantly in Family Health Teams (75% vs 19.5%).

### 3.4. Number of Patients on Opioids

The 24 physicians in the ECHO group prescribed opioids to 2760 patients during the study period, the 96 physicians in the control group prescribed to 11,117 patients, and the 3000 Ontario family physicians prescribed to 374,174 patients.

The number of patients per physician taking opioids was not significantly different across the groups in any time point (*p*-values varied from 0.492 to 0.94). During the period of 12 to 6 months before ECHO, the ECHO participants had an average of 79 patients on opioids compared to 78 and 83 of the matched cohort and Ontario group, respectively, and this difference was not statistically significant (*p* = 0.81). Twelve months after ECHO, the average number per group was 79, 81 and 80, respectively (*p* = 0.98).

### 3.5. Proportion of Patients on High-Dose Opioids

The proportion of patients on high-dose opioids was significantly different across groups. (Figure 1). During the period of 12 to 6 months before ECHO, the ECHO participants had 11% of patients on a high dose of opioids compared to 8% of the matched cohort and 7% of the Ontario sample, and this difference was statistically significant (*p* = 0.014). In the period of six to 12 months post-ECHO, the proportion per group was 10%, 9%, and 7%, respectively, and this difference was statistically significant (*p* = 0.01).

The six month pre/post change in the number of patients on a high dose of opioids was not different among the groups (*p* = 0.5); however, the 12 month pre/post change showed a statistically significantly change, with a small effect size of 0.26 between ECHO and the matched group and 0.2 between ECHO and the Ontario group (*p* = 0.028).

Over time, the change in the mean proportion of patients on a high dose did not differ significantly at six or 12 months pre/post ECHO for the matched cohort or the Ontario group; however, the ECHO group at 12 months pre/post was the only group with a reduction in the proportion of patients on a high dose (from 11% to 10%).

### 3.6. Proportion of Patients on Fentanyl

During the period of 12 to six months pre-ECHO, the ECHO group had a higher number of patients on fentanyl, with a mean of 3.88 patients per physician, compared to 3.57 in the matched cohort and 2.57 in the Ontario group (*p* = 0.031), and there was a drop in this number in the post-ECHO period for all three groups, to 2.50 in the ECHO group, 2.83 in the matched cohort and 2.07 in the Ontario group (*p* = 0.1). We also observed that the proportion of fentanyl prescriptions among patients taking a high dose of opioids decreased over time in all three groups: 34% to 23% in the ECHO group, 37% to 28% in the matched cohort, and 31% to 27% in the Ontario group.

There were no significant differences at six or 12 months pre/post ECHO among the three groups. However, all groups had a reduction, and the reductions in the ECHO group occurred more at the six months pre/post period, while the changes in the matched cohort and the Ontario group happened during the longer timeframe of the 12 months pre/post periods.

### 3.7. Daily Morphine Equivalent Doses for all Patients on Opioids

At the six months pre-index date, the ECHO group was prescribing approximately 50% higher doses, measured in daily MEDs, than the matched control group and the Ontario physicians (Figure 2). The matched control group and the Ontario physicians were prescribing approximately 70 mg of MEDs daily, while the ECHO physicians were prescribing approximately 110 mg of MEDs daily. Throughout the 12 months of observation, the ECHO group showed a trend toward lower doses, while both the matched control and Ontario groups remained stable at almost the same dose of 66 to 73 mg of MEDs daily. The ECHO group initiated reducing MEDs a few months before they attended ECHO. After six months in the ECHO group, the average MED was 88 mg daily, and this was a statistically significant reduction (*p* = 0.0047). At six months post-ECHO, the ECHO group was still above the average of the matched cohort and the Ontario physicians, but this difference was not significant. The pre- and post-ECHO in average daily MED for the matched group were 71 and 70, respectively, and not statistically significantly different. But the pre/post daily MED for the Ontario group were 69 and 66 mg, respectively, and this difference was statistically significant (*p* = 0.0001).

The percent reduction in the daily MEDs was 19.8% in the ECHO group, 1.8% in the matched cohort and 4.5% in the Ontario group when considering the six months pre to six months post periods.

### 3.8. Daily Morphine Equivalent Doses for Patients on High dose Opioids (above 200 mg MED Daily)

When we looked at only the group of patients who were receiving a daily MED above 200 mg we observed that, at baseline, the ECHO group was prescribing a much higher average daily dose of opioids (350 mg MED daily) than the matched control group (296 mg MED daily) and the Ontario group (302 mg MED daily) (Figure 3). The ECHO group started lowering the dose of opioids a few months before they attended ECHO, and they maintained the reduction up to six months after. The ECHO group crossed to levels below the matched controls and the Ontario physicians at six months post-ECHO evaluations. This reduction in mean daily MED from 350 to 279 mg/day was statistically significant (*p* = 0.01) among the ECHO group. There was no significant change in daily MED for the matched control group with the pre six months value at 296 mg/day and the post at 298 mg/day. Among the Ontario controls, there was a small reduction from 302 mg/day at pre six months to 297 mg/day post six months, and this was also not statistically significant (*p* = 0.22).

The percent reduction in the daily MEDs was 20.2% in the ECHO group (*p* = 0.01), −0.8% (*p* > 0.05) in the matched cohort, and 1.5% (*p* > 0.05) in the Ontario group for patients on a high dose of opioids when considering the six months before and six months after periods.

## 4. Discussion

Using administrative database, this study was the first in Canada to suggest that participation in a tele-mentoring program such as ECHO might change physicians’ opioid prescribing behaviours. Objective measures of behavioural changes have been difficult to collect due to the lack of comparison groups. In our study, we were able to not only have an objective measure of opioid prescribing, but also to create two comparison groups. The matched control group was an attempt to create a group with similar characteristics to the ECHO group, much like a randomized trial. The Ontario group was important to show that there were no regional or temporal trends in opioid prescribing in Ontario during this study period. A similar study was recently published in the US Army and Navy ECHO Pain Telementoring program. Katzman et al. studied 99 clinics who attended ECHO and compared them to 1283 clinics that did not participate in ECHO. The clinics that attended ECHO had a greater percent decline than the comparison clinics in the number of opioid users, annual opioid prescriptions per patient, average MED prescribed per patient per year, days of co-prescribed opioid and benzodiazepine per opioid user per year [8].

The physicians who attended ECHO and were included in this study were mostly women, working or living in urban settings and working in family health teams in many remote geographical areas of Ontario. The ECHO physicians who did not agree to participate were not significantly different from those who agreed to participate in this study. We found that the ECHO group and the two control groups had a very similar average number of patients on opioids, and that the average numbers did not vary significantly over time. The ECHO physicians were prescribing much higher opioid doses to their patients compared to the other two control groups, with approximately 50% higher doses during the period of six months prior to their starting in ECHO. There was a reduction in high-dose opioid prescriptions in the ECHO group, but not in the control groups during the same comparable periods. It is uncertain if the changes observed during the study period will be sustained over a longer period. Ideally, a longitudinal study with a longer follow-up period is needed to fully understand the benefits of the ECHO program.

One of the explanations as to why the ECHO physicians were prescribing higher doses of opioids to their patients is possibly because those physicians self-selected to participate in ECHO. They might have a perception that they are prescribing more than their colleagues, and therefore entered ECHO as an opportunity to learn more about management of chronic pain. This indicates that ECHO is achieving its goals of attracting physicians who are prescribing high-dose opioids and mentoring them on how to manage pain and taper opioids appropriately. It was also reassuring to see that the number of patients on opioids remained stable during the study period among the ECHO physicians, probably an indication that they were not firing or discharging patients on opioids, which could be a measure to avoid the problem of managing complex patients on opioids. In ECHO sessions, opioid tapering is taught during the didactics and case discussions, and the emphasis is always on the appropriateness of taper, using methods that are compassionate, responsible and slow.

A limitation of the current study was that the NMS database contains records of all opioids dispensed in Ontario, which includes any opioid dispensed for acute pain, chronic pain, opioid use disorders, or cancer. Therefore, we did not have diagnostic information to identify the reasons for opioid prescription; examining the health characteristics of patients receiving opioids is feasible using administrative databases but would only be able to provide information about the patient’s profile, not the particular reason for opioid prescription. Examining patient characteristics was outside the scope of this study. We can assume that acute pain was not included in the high-dose opioid analyses because the threshold of 200 mg MED would be too high to treat acute or post-operative pain [18]. Further, the NMS is a record of opioids dispensed; it was assumed for the purposes of this study that patients were taking the described dose starting on the date the opioid was dispensed.

Although we obtained a good response rate from physicians (38% from 26 consented out of 67 eligible), our conclusions are based on a small sample of 24 physicians from ECHO. We cannot exclude the possibility that physicians who were not included in the analyses refused to participate due to irresponsible or inappropriate opioid prescriptions, or did not want to jeopardize the ECHO program through their opioid prescriptions.

Another important limitation of this study is that we cannot attribute all the results to participation in ECHO alone. It is possible that the 24 physicians who were included in the analyses were also learning from other sources or making other changes to their practices that were outside the scope of ECHO.

Finally, we do not know if the reduction in opioid doses reflects any improvement in patients’ quality of life, pain or function. It is possible that some patients receiving fewer opioids after ECHO did not have a good experience with the opioid tapering, and therefore they may return to the original dose of opioids, either by the same prescriber, a different prescriber, or buying from illicit sources.

Ideally, the best study design to test if participation in ECHO leads to a change in opioid prescribing would be a randomized trial where family physicians interested in ECHO would be randomized to attend ECHO immediately, or to a control group (waitlist or a different medical education). The problem with this design is that bias could be introduced if waitlisted physicians implement practice changes due to regulations or policies that are outside of the scope of ECHO.

Studying the effectiveness of ECHO on patients’ outcomes is highly desirable. Patients who are treated by physicians who attended ECHO are recommended to be included in quantitative and qualitative research. The impact of ECHO knowledge on patients in those providers’ practices is unknown. Future studies should examine use of walk-in clinics and emergency room use, as well as quality of life and pain levels of patients whose opioid dose has been reduced by ECHO physicians.

## 5. Conclusions

ECHO participants did not show a reduction in the proportion of patients on opioids before and after ECHO. However, they showed a significant reduction in the daily doses of opioids, especially among patients who were on high-dose opioids, a reduction that was not seen in the matched cohort or the random sample of Ontario family physicians.

## Figures and Tables

**Figure 1 jcm-09-00014-f001:**
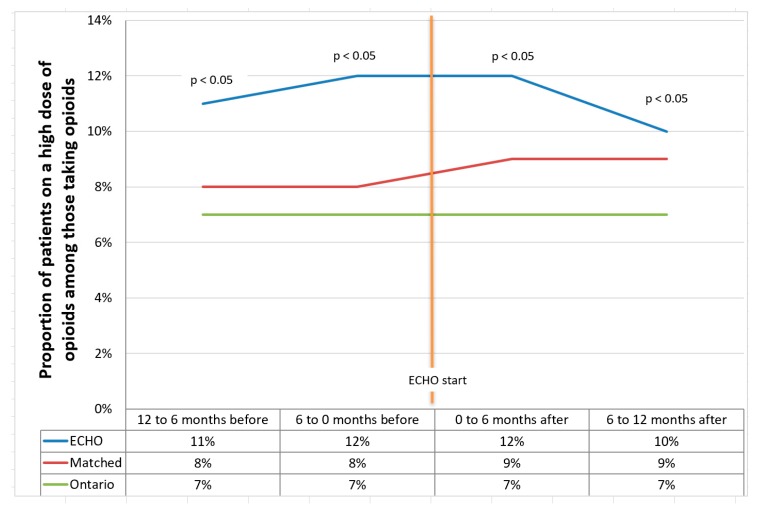
Proportion of patients on a high dose of opioids before and after extension for community health outcomes (ECHO) start.

**Figure 2 jcm-09-00014-f002:**
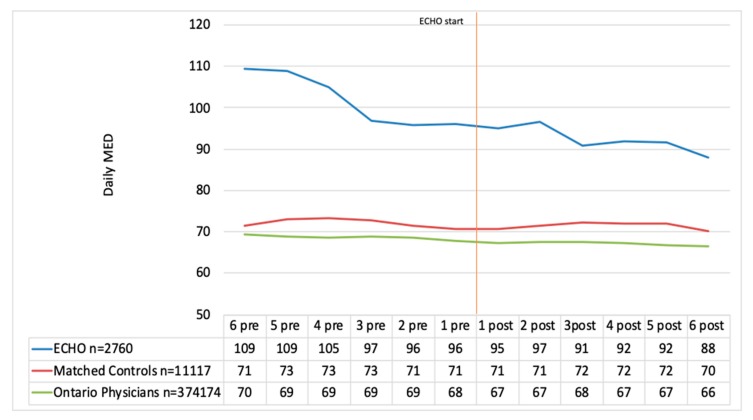
Daily morphine equivalent dose (MED) of all patients on opioids (monthly average) before and after ECHO.

**Figure 3 jcm-09-00014-f003:**
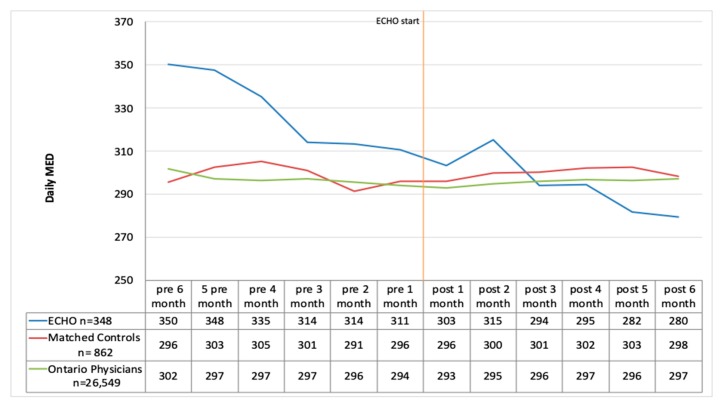
Daily morphine equivalent dose (MED) for patients on a high dose of opioids (monthly average) before and after ECHO.

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
