# Peer review of "Changes in Opioid Prescribing Behaviors among Family Physicians Who Participated in a Weekly Tele-Mentoring Program"

_jcm, 2019, doi:10.3390/jcm9010014_

Round 1

Reviewer 1 Report

This is a concise and interesting report.

It is well known that opioid abuse has become a social problem in Canada.

In other countries, such as Japan, it should be used as a reference for future opioid measures.

This paper is important for understanding the current situation.

The tabulation is easy to understand, which is useful for readers.

Author Response

Thank you for your review of our manuscript.

Reviewer 2 Report

Dear authors,

Thanks for submitting your work to the journal. You describe how a tele-mentoring program called ECHO can be associated to a decrease of opioid prescription, and especially at high doses. You used a retrospective design, focusing on administrative prescription data, and matched family physicians with controls.

Interestingly, the physicians attending the ECHO program prescribed more opioids before the program than the controls. They tended to approach the controls after the program, but the decrease seems to have started before. Altogether, this is very encouraging, as suggesting that the problem of high doses of opioids may be managed by sensitising and supporting the prescribers. Methodologically, this is challenging as suggesting that the ECHO participants are not similar to the controls, because prescribing more but already sensitized to the need for a decrease in the opioids prescriptions.

The authors should point as limitation the relative low attendance rate to the ECHO sessions, and consider the potential interference of the attendance rate with the decrease of opioid prescriptions. Less sessions may be associated with less decrease.

More important, I would suggest to the authors to choose one clearly identified primary outcome, and to consider all the others as secondary. These could be mentioned more briefly, ideally in figures or tables, with details in appendix. This depending on the editor advice.

Finally, the perspective of a randomised trial should not be pushed away. If the changes in policies neutralise the effect of ECHO, that's great. If not, ECHO could be an additional resources, needing investment, but potentially cost effective for the community.

Reviewer 3 Report

The manuscript is well written, thoughtful, and thoroughly explains the context for the study setting in Ontario, Canada. The limitations of the study are well described in the Discussion section. The study design is sound and the analyses are appropriate. Some suggestions for improvement are described below:

Although the study references the period of June 2014 - December 2016 and a data collection period of 12 months pre- to 12 months post ECHO intervention, it is unclear whether ECHO is an ongoing program after 2016 available to providers and whether there was any educational follow-up to prescribers and their institutions about their prescribing patterns that were uncovered during the study period. These points should be clarified in the manuscript.

It would be helpful to know the average amounts of experience/years in practice that each cohort had in prescribing opioids. The text says that years in practice were similar among the groups but this additional information could help readers to understand at what time periods the providers were medically trained and what was medical and public opinion and knowledge of opioid use during that time.

In section 2.7 Data Analyses more information is needed on how the matched controls ratios were designed e.g. 1:1 vs. 3:1 vs 4:1 etc., how confounding factors were accounted for in the matching, and how validity and precision were determined.

In the Discussion section line 302, more explanation could be provided regarding why the ECHO physicians were prescribing higher doses of opioids to their patients such as factors related to the institution or medical insurance.

It would be beneficial to discuss whether or not the ECHO program is expected to have long term sustainability in changing physician practices in opioid prescribing.

It appears that a longitudinal study with a longer follow-up period is needed to fully understand the benefits of the ECHO program and this point can be discussed. 

In Figure 1, the p-values should be clearly indicated on the graph as p < 0.05. Also the word "months" should be included in the table to describe 12 to 6 before etc.

In the text the figures should be consistently labeled as Figure 1 and Figure 2.
